

# Digging for DNA at depth: rapid universal metabarcoding surveys (RUMS) as a tool to detect coral reef biodiversity across a depth gradient

Joseph D. DiBattista[1,2], James D. Reimer[3,4], Michael Stat[1,5],
Giovanni D. Masucci[3], Piera Biondi[3], Maarten De Brauwer[1] and
Michael Bunce[1]

[1] Trace and Environmental DNA (TrEnD) laboratory, School of Molecular and Life Sciences, Curtin University of Technology, Perth, WA, Australia
[2] Australian Museum Research Institute, Australian Museum, Sydney, NSW, Australia
[3] Graduate School of Engineering and Science, University of the Ryukyus, Okinawa, Japan
[4] Tropical Biosphere Research Center, University of the Ryukyus, Okinawa, Japan
[5] Department of Biological Sciences, Macquarie University, North Ryde, NSW, Australia

Corresponding author
Joseph D. DiBattista,
josephdibattista@gmail.com

## ABSTRACT

**Background:** Effective biodiversity monitoring is fundamental in tracking changes in ecosystems as it relates to commercial, recreational, and conservation interests. Current approaches to survey coral reef ecosystems center on the use of indicator species and repeat surveying at specific sites. However, such approaches are often limited by the narrow snapshot of total marine biodiversity that they describe and are thus hindered in their ability to contribute to holistic ecosystem-based monitoring. In tandem, environmental DNA (eDNA) and next-generation sequencing metabarcoding methods provide a new opportunity to rapidly assess the presence of a broad spectrum of eukaryotic organisms within our oceans, ranging from microbes to macrofauna.

**Methods:** We here investigate the potential for rapid universal metabarcoding surveys (RUMS) of eDNA in sediment samples to provide snapshots of eukaryotic subtropical biodiversity along a depth gradient at two coral reefs in Okinawa, Japan based on 18S rRNA.

**Results:** Using 18S rRNA metabarcoding, we found that there were significant separations in eukaryotic community assemblages (at the family level) detected in sediments when compared across different depths ranging from 10 to 40 m ($p = 0.001$). Significant depth zonation was observed across operational taxonomic units assigned to the class Demospongiae (sponges), the most diverse class (contributing 81% of species) within the phylum Porifera; the oldest metazoan phylum on the planet. However, zonation was not observed across the class Anthozoa (i.e., anemones, stony corals, soft corals, and octocorals), suggesting that the former may serve as a better source of indicator species based on sampling over fine spatial scales and using this universal assay. Furthermore, despite their abundance on the examined coral reefs, we did not detect any octocoral DNA, which may be due to low cellular shedding rates, assay sensitivities, or primer biases.

**Discussion:** Overall, our pilot study demonstrates the importance of exploring depth effects in eDNA and suggest that RUMS may be applied to provide a baseline of information on eukaryotic marine taxa at coastal sites of economic and conservation importance.

## INTRODUCTION

In coral reef ecosystems, shifts in community structure often occur at small spatial scales. For example, marine taxa may be restricted to specific reef zones (e.g., lagoon, reef crest, fore reef; *Menza, Kendall & Hile, 2008*), or separated by depth (*Friedlander & Parrish, 1998*; *Kahng & Kelley, 2007*; *Brokovich et al., 2008*), which represents the steepest environmental gradient on coral reefs. Increasing depth is associated with decreases in light irradiance, wave action, nutrients, and temperature variation (*Lesser, Slattery & Leichter, 2009a*; *Slattery et al., 2011*). Reef-building corals and other anthozoans in particular show pronounced variation in morphology (*Nir et al., 2011*) and in the composition of their symbiotic Symbiodiniaceae (*Lesser et al., 2009b*; *Bongaerts et al., 2013*; *Kamezaki et al., 2013*) across depth gradients. Coral reef fish communities similarly exhibit changes in species richness and composition with depth (*Brokovich et al., 2008*; *Bejarano, Appeldoorn & Nemeth, 2014*).

Until recently, spatial surveys of marine biodiversity have primarily focused on megafauna and macrofauna (*Gaston, 2000*; *Tittensor et al., 2010*) or microfauna (*Sunagawa et al., 2015*; *Soliman et al., 2017*), rather than meiofauna (the polyphyletic group of organisms that fall somewhere in between) (*Lambshead & Boucher, 2003*; *Giere, 2008*; *Fonseca et al., 2010*; *Curini-Galletti et al., 2012*; *Fonseca et al., 2014*; *Guardiola et al., 2015*; *Leray & Knowlton, 2015*; *Guardiola et al., 2016*). These organisms arguably represent the most abundant component amongst benthic metazoans in all marine systems from the intertidal zone to the deep-sea floor (*Danovaro & Fraschetti, 2002*; *Giere, 2008*). A major bottleneck in meiofaunal surveys is related to the time and expertise required for the analyses of distinctive morphological characters. This taxonomic limitation can now be largely overcome with a combination of environmental DNA (eDNA) and next-generation sequencing metabarcoding, which offers a rapidly developing avenue to assess the presence of a broad spectrum of eukaryotic organisms within our oceans (*Kelly et al., 2017*; *Ransome et al., 2017*; *Stat et al., 2017*).

Environmental DNA has been defined by *Taberlet et al. (2018)* as "a complex mixture of genomic DNA from many different organisms found in environmental samples," a definition which includes bulk samples of water, air, sediment, or plankton. eDNA recovered from complex multi-species substrates are often now combined with metabarcoding approaches, defined by *Taberlet et al. (2012)* as "high-throughput multispecies (or higher-level taxon) identification using the total and typically degraded DNA extracted from an environmental sample." This approach can now provide a

cost-effective and rapid assessment of biodiversity localized to individual coral reefs (*Stat et al., 2018*). Previous studies have focused on a range of organisms, from unicellular eukaryotes (i.e., protists) (*De Vargas et al., 2015*) to large animals (*Bakker et al., 2017*), thought to be detected via the capture of DNA fragments or whole cells shed from the target organism. Benthic collection methods (i.e., Autonomous reef monitoring structures (ARMS)) combined with metabarcoding using universal primer sets have also proven useful in surveying cryptobenthic biodiversity not revealed by visual techniques (*Al-Rshaidat et al., 2016*; *Pearman et al., 2016*, *2018*). ARMS and comparable methods, however, are not without their own taxonomic biases (*Ransome et al., 2017*), need to be deployed for months to years in order for sufficient animals to settle in the fibrous matrix, and often require taxonomic specialists to identify the larger fraction of organisms (*Pearman et al., 2016*). A lack of reference DNA sequences for many marine taxa further hinders their identification here and in other applications.

In this pilot study, we test whether sampling of marine sediment combined with eDNA metabarcoding using universal 18S rRNA primers can provide reliable information about the broad spectrum of taxonomic diversity (at the family level) stratified by depth along subtropical coral reefs. Sediment was selected as the biological substrate as ongoing work suggests that it reveals more benthic families compared to seawater sampling (*Koziol et al., in press*). We also tested whether taxonomic families of interest were specialized to specific depths, with a focus on the classes Anthozoa (phylum Cnidaria) and Demospongiae (phylum Porifera). Anthozoa includes anemones, stony corals, soft and octocorals, whereas Demospongiae (sponges) encompasses 81% of all sponge species (*Van Soest et al., 2017*). Contrary to popular belief, on tropical and subtropical reefs, sponge diversity can in fact be higher than that of corals (*Diaz & Rützler, 2001*), although their taxonomy is not yet resolved. Both of these groups play an important role in the functioning of coral reef ecosystems, such as recycling dissolved organic matter (*Rix et al., 2016*). For example, sponges on coral reefs absorb dissolved organic carbon and return it to the reef via particulate detritus, otherwise known as the "sponge loop" (*De Goeij et al., 2013*).

We chose to focus our efforts on the coastal marine ecosystems of Okinawa, Japan, which are recognized for their high levels of biodiversity and endemism (*Roberts et al., 2002*). This coastline faces growing anthropogenic pressures due to increased coastal development (*Reimer et al., 2015*; *Heery et al., 2018*), as well as terrestrial input in the form of pollutants (*Ramos, Inoue & Ohde, 2004*; *Imo et al., 2008*) and nutrient runoff (*Shilla et al., 2013*). Moreover, the coral reefs of Okinawa have been subject to the effects of climate change, with extreme coral bleaching occurring during the 1998 El Niño-Southern Oscillation (*Tsuchiya et al., 2004*) and more recently in 2015–2017 (*Kayanne, Suzuki & Liu, 2017*; *Ministry of the Environment, 2017*). Current coral reef biodiversity monitoring efforts in Japan are generally limited to scleractinian corals (i.e., stony corals or hard corals) and fish, and from these data the overall trend for coral reefs in Okinawa is that of an ecosystem in decline (*Hongo & Yamano, 2013*). Here, we examine the potential for universal metabarcoding surveys (rapid universal metabarcoding surveys (RUMS)) of eDNA in sediment samples to provide snapshots of marine biodiversity that can serve as a baseline to be revisited at future points in time.

## MATERIALS AND METHODS

### Sampling sites

The coral reef sites in Okinawa, Japan that we selected were minimally impacted by natural (no freshwater input) and anthropogenic disturbances (no coastal development), although the presence of discarded fishing line at both dive sites suggests some recreational fishing pressure. Cape Hedo, Kunigami (26.87228°N, 128.26652°E) is the northernmost point of the main island of Okinawa-jima and is topographically complex, with more than 50% hard coral cover at shallower sites (<20 m), and an abundance of sponges and octocorals due to consistently fast currents, which can be seen on the northeast coast (*Kudaka et al., 2008*). Rukan Reef (26.09961°N, 127.53962°E) is a small submerged atoll ~10 km offshore to the southwest of Okinawa-jima, and includes a small lagoon with no land above sea level other than a lighthouse set on concrete blocks. The reef at Rukan is also subject to strong oceanic currents, with a diverse coral community (*Ohde & Van Woesik, 1999*), including large octocorals and sponges on the reef slopes. Both sites are characterized by steep drop-offs with walls greater than 45° sloping downward; the Rukan reef wall ends at ~30 m depth and levels off into a coral rubble field, whereas the Cape Hedo reef continues to drop down to depths greater than 100 m.

### Sediment collections

Four replicates of approximately 10 g of marine sediment were collected with sterile 15 ml falcon tubes at 10, 20, and 30 m depth at two reefs (Cape Hedo and Rukan), as well as 40 m depth at one reef (Cape Hedo) in July 2016 in Okinawa, Japan on SCUBA (Fig. 1). The sterile falcon tubes remained closed on the dive until the moment of sampling, and were closed immediately after scooping up the sample; each diver was cautious not to touch the inside of the lid or tube with their own hands, and the diver performing sampling wore gloves. We chose to focus on surface sediment (<5 cm below the substrate) given that their eDNA concentrations can be higher than those in surface seawater (*Torti, Lever & Jørgensen, 2015*) and appear to yield a greater fraction of benthic diversity (*Koziol et al., in press*). Sediment sampling was repeated at only one of the reefs (Cape Hedo) at 10, 20, 30, and 40 m depths in October 2017. Four replicate samples were collected for each site, depth, and year. Sediment samples were placed on ice in a cooler within sterile plastic bags and then frozen at −20 °C until processing in a dedicated PCR-free DNA extraction laboratory at Curtin University in Perth, Australia.

### DNA extraction

Total nucleic acids were extracted from each replicate sediment sample following homogenization of 0.5–0.8 g of organic material using bead tubes mixed on a Minilys® homogenization machine (Bertin Technologies, Aix-en-Provence, France). Homogenized replicates were transferred into sterile two ml microfuge tubes. Although singly subsampling ~0.5 g from a much larger volume of sediment (i.e., ~10 ml) may have missed variance within the sample, grouping of replicates at each site provided confidence in the site-specificity of the community assemblage that we detected with this experimental design.

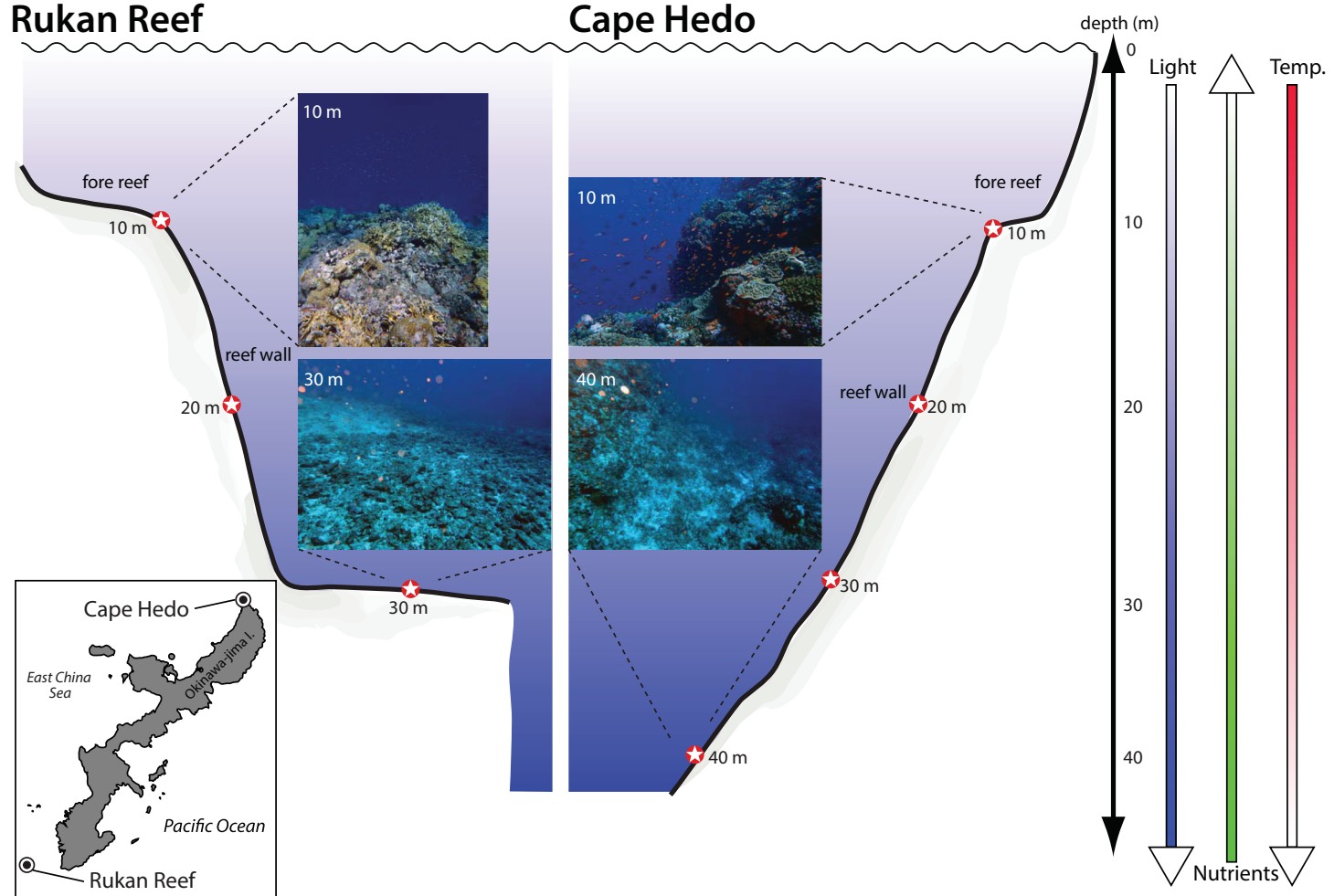

**Figure 1 Location and depth of sediment samples collected at two coral reefs in Okinawa, Japan.** Location and depth of sediment samples collected at two coral reefs in Okinawa, Japan. Photographs provide representative views of the substrate for each location at the minimum and maximum depth sampled. Shaded arrows indicate the direction of depth gradients related to light penetration, nutrients, and water temperature (temp.).

DNA extraction was completed using the MoBio Powersoil extraction kit (MoBio Laboratories, Carlsbad, CA, USA) following the manufacturer's protocol, with a modification in the reaction volume of the homogenate to double the default quantity. Purified DNA was then eluted into a final volume of 100 μl. Four DNA extraction controls were also included in the workflow, which were processed along with experimental samples in the same manner, save for the absence of sediment. This kit was chosen because of the advantage of co-purification of inhibitors in sediment samples.

## DNA amplification

A universal primer set targeting 18S rRNA (V1-3 hypervariable region; 18S_uni_1F: 5′—GCCAGTAGTCATATGCTTGTCT—3′; 18S_uni_400R: 5′—GCCTGCTG CCTTCCTT—3′; *Pochon et al., 2013*) with an amplicon length of ~340–420 bp was used to maximize the eukaryotic fraction of marine diversity detected along a coral reef depth

gradient. Quantitative PCR (qPCR) experiments were set up in a dedicated ultra-clean laboratory at Curtin University designed for ancient DNA work using a QIAgility robotics platform (Qiagen Inc., Valencia, CA, USA). Given that low copy number and PCR inhibition can severely impact metabarcoding data (*Murray, Coghlan & Bunce, 2015*), template input concentrations were optimized using a qPCR dilution series (neat, 1:10, 1:100) based on the reaction conditions described below. To reduce the likelihood of cross-contamination, chimera production, and index-tag jumping (*Esling, Lejzerowicz & Pawlowski, 2015*), amplification of target DNA was performed in a single round of PCR using fusion-tag primers consisting of the 18S primers coupled to Illumina adaptors, custom sequencing primers, and index combinations unique to this study. All qPCR reactions for each replicate were run in duplicate and subsequently pooled to control for amplification stochasticity. PCR reagents included 1 × AmpliTaq Gold® Buffer (Life Technologies, Carlsbad, CA, USA), two mM $MgCl_2$, 0.25 μM dNTPs, 10 μg BSA, five pmol of each primer, 0.12 × SYBR® Green (Life Technologies, Carlsbad, CA, USA), one Unit AmpliTaq Gold DNA polymerase (Life Technologies, Carlsbad, CA, USA), two μl of DNA, and Ultrapure™ Distilled Water (Life Technologies, Carlsbad, CA, USA) made up to 25 μl total volume. PCR was performed on a StepOnePlus Real-Time PCR System (Applied Biosystems, Foster City, CA, USA) under the following conditions: initial denaturation at 95 °C for 5 min, followed by 45 cycles of 30 s at 95 °C, 30 s at 52 °C, and 45 s at 72 °C, with a final extension for 10 min at 72 °C.

## DNA sequencing

Libraries for sequencing were made by pooling amplicons into equimolar ratios based on qPCR $C_T$ values and the endpoint of amplification curves. Amplicons in each pooled library were size-selected using a Pippin Prep (Sage Science, Beverly, MA, USA) and purified using the Qiaquick PCR Purification Kit (Qiagen Inc., Valencia, CA, USA). The volume of purified library added to the sequencing run was determined against DNA standards of known molarity on a LabChip GX Touch (PerkinElmer Health Sciences, Waltham, MA, USA). Final libraries were sequenced paired-end using a 500 cycle MiSeq® V2 Reagent Kit and standard flow cell on an Illumina MiSeq platform (Illumina, San Diego, CA, USA) located in the Trace and Environmental DNA Laboratory at Curtin University. These samples were included in a mixed run with additional samples from a related study, and therefore did not receive the full output of sequence reads from the standard kit.

## Bioinformatic filtering

All sequence data were quality filtered (QF) prior to taxonomic assignment and operational taxonomic units (OTU) analysis. Metabarcoding reads recovered by paired-end sequencing were first stitched together using the Illumina MiSeq Reporter software under the default settings. Sequences were then assigned to samples based on their unique index combinations and trimmed in Geneious® Pro v 4.8.4 (*Drummond et al., 2009*). In order to eliminate low quality sequences, only those with 100% identity matches to Illumina adaptors, index barcodes, and template specific oligonucleotides were kept for downstream analyses. Sequences were further

processed in USEARCH v 9.2 (*Edgar, 2010*). This program was used to trim ambiguous bases, remove sequences with average error rates >1%, remove sequences <200 base pairs, dereplicate each sample, abundance filter unique sequences using both conservative ($\geq$5 identical reads) and less conservative ($\geq$2 identical reads) thresholds, and remove chimeras. Given that both the conservative and less conservative QF workflow gave comparable results (raw data available from Dryad Digital Repository https://doi.org/10.5061/dryad.37qv5rd), we only present data where a minimum of two identical reads are required as a threshold. It should also be noted that the sequences in each replicate were sub-sampled to 20,000 sequences prior to dereplication to ensure that sampling effort was even among replicates; a random subset of sample species accumulation curves are included in Fig. S1. We found that 20,000 sequences struck a balance between the inclusion of samples and the detection of families within each replicate. Despite modest amplification, none of the extraction controls retained sequences following the QF pipeline and are therefore not reported further.

## Taxonomic assignment

Unique sequences that passed QF were queried against the National Center for Biotechnology Information (NCBI) nucleotide database using BLASTn on the Magnus Cray XC40 system located at the Pawsey Supercomputing Centre in Perth. The BLASTn settings were as follows: -num_alignments 25; -num_descriptions 25; -reward 1; -qcov_hsp_perc 100; -perc_identity 90. BLASTn results were imported into MEtaGenome ANalyzer (MEGAN) v 5.11.3 (*Huson & Weber, 2013*) and taxonomic identities assigned at the family level based on the lowest common ancestor algorithm (minimum bit score = 600; top percent of reads = 5%; max expected = 0.01). Rarefaction analyses were performed in MEGAN (see Fig. S1) and all taxonomic nomenclature was based on the World Register of Marine Species (*Van Soest et al., 2017*).

Given the lack of reference barcodes for many marine taxa, OTUs were also identified for all QF 18S metabarcoding data assigned to the classes Anthozoa (i.e., anemones, stony corals, soft corals, zoantharians, antipatharians, and naked corals) and Demospongiae (i.e., sponges) in MEGAN using the "extract reads" function; this provided a taxonomy-independent comparison for these groups that could effectively be aligned. This process followed the MiSeq SOP outlined in *Kozich et al. (2013)*. OTUs were parsed using a 99% sequence similarity in Mothur v 1.35.1 (*Schloss et al., 2009*). A 99% OTU threshold was used for both classes as this represents a conservative cut-off between different Anthozoa (*Shearer et al., 2002*; *Huang et al., 2008*) (but not Ceriantharia; *Stampar et al., 2014*) and Demospongiae (*Redmond et al., 2013*; A. Collins, 2017, personal communication) species in order to avoid unnecessarily splitting OTUs. Representative sequences from each OTU were then compared against previously reported GenBank sequences using BLASTn (*Altschul et al., 1990*) for further identification.

## Statistical analyses

Family richness was calculated per sample based on the taxonomic composition of marine eukaryotes identified by the 18S gene and analyzed using the R software package

(*R Development Core Team, 2015*). Kruskal–Wallis tests were used to compare taxonomic richness between depths as data did not meet assumptions of normality.

Taxonomic composition of marine eukaryotes at the family level for 18S was analyzed using PRIMER v 7 (*Clarke & Gorley, 2015*). Data were presence/absence transformed and a Jaccard resemblance matrix was constructed to assess the effect of depth on biological community assemblages. Differences among depths was tested using PERMANOVA (One factor design: Depth (Fixed)) under a reduced model with 9,999 permutations. Pairwise PERMANOVA tests were conducted to compare different depths. Canonical analysis of principal coordinates (CAP) was used to visualize differences among categories. Leave-one-out allocation success tests were used to estimate misclassification errors and test the uniqueness of assemblages (*Anderson & Willis, 2003*). Plots were overlaid with vectors of the taxa most closely correlated with figure axes (Pearson's correlation value > ±0.4). This entire process was repeated for the combined taxonomy-independent (i.e., OTU) metabarcoding data for classes Anthozoa and Demospongiae.

## RESULTS

Using a universal metabarcoding assay targeting the 18S rRNA gene, a total of 3,787,288 amplicon reads were sequenced from 42 samples to provide a snapshot of eukaryotic biodiversity along a depth gradient at two coral reefs in Okinawa, Japan (Table S1). All 42 samples amplified, but two did not pass the QF thresholds for inclusion in the statistical analysis (AWFS_F16_0429, Cape Hedo, 20 m, 2016; SED126, Cape Hedo, 20 m, 2017). The mean number of sequences per sample was 90,174 ± 84,764 SD (Table S1). The metabarcoding data was assigned to 85 eukaryotic classes, 149 orders, and 222 families (Table S2). These included a number of reef-forming benthic organisms, including coralline red algae (Class Florideophyceae), polychaete worms (class Polychaeta), tunicates (class Ascidiacea), bivalves (class Bivalvia), a variety of hexacorals (class Anthozoa), calcareous sponges (class Calcarea), and demosponges (class Demospongiae) (for summary see Fig. 2). On average, 440 ± 223 SD unique sequences were assigned per sample, whereas, on average, 881 ± 278 SD unique sequences remained unassigned (Table S1), which justified additional downstream taxonomy-independent analyses using OTUs.

Taxonomic diversity based on family richness was not significantly different across depths ($p = 0.79$, d$f = 3$, $\chi^2 = 1.01$; Fig. 3A), but PERMANOVA tests revealed significant differences in marine community assemblages among the different depths ($p = 0.01$, d$f = 3$, pseudo-F = 1.28). The significant differences for depth were between 10–20 m and 10–30 m (Data S1). Based on a Venn diagram, there was modest overlap in families shared between depths compared to families unique to specific depths (Fig. 3B).

Constrained CAP analysis supported the notion that there was minimal overlap between marine community assemblages at different depths, from both sites, with the exception of between 20 and 30 m (Fig. 4). The allocation success for different depths was 57.5% overall (Trace statistic: 2.39; $p < 0.001$), with the highest assignment at

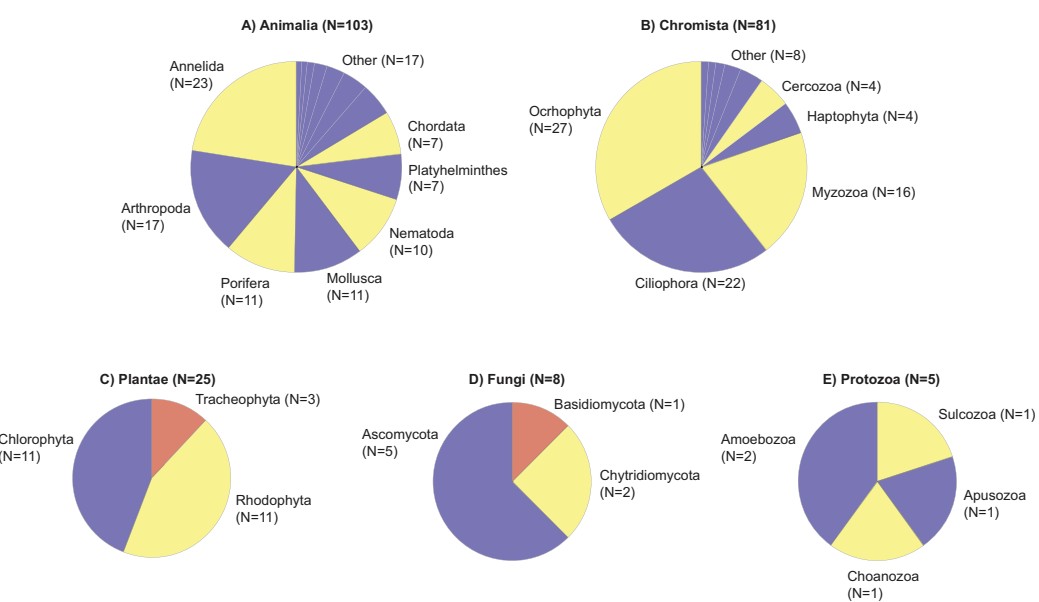

**Figure 2 Taxonomic phylogram of eukaryotic diversity based on sediment samples collected at two coral reefs in Okinawa, Japan and 18S rRNA sequences.** Taxonomic phylogram of eukaryotic diversity based on sediment samples collected at two coral reefs in Okinawa, Japan and 18S rRNA sequences. Pie segments (A–E) indicate the phyla detected within each kingdom, with the number of families detected within each phyla indicated in parentheses. Color is used only to provide contrast between adjacent pie segments. "Other" represents the number of families in a phyla that make up <5% of the total number of families detected in that Kingdom.

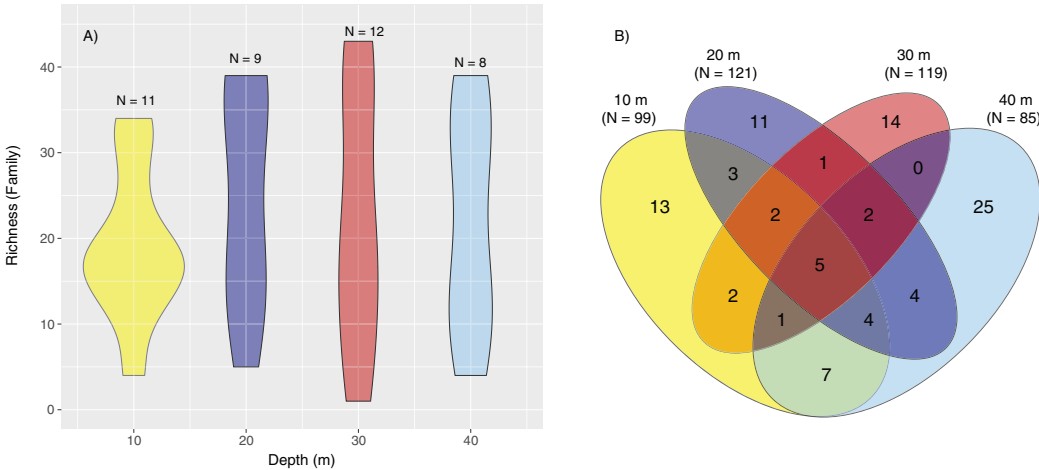

**Figure 3 Violin plot representing family richness and Venn diagram representing the number of families identified by depth.** Violin plot representing family richness (A) estimates based on sediment samples collected at two coral reefs in Okinawa, Japan and 18S rRNA sequences. The Venn diagram (B) represents the number of families identified by depth. Yellow, dark blue, red, and light blue segments of the Venn diagram represent the number of families identified at 10, 20, 30, and 40 m, respectively, with shaded colors indicating the shared number of families across different depths.

10 m (72.7%), followed by 30 m (58.3%), 40 m (50%), and then 20 m (44.4%). This differential allocation success further confirms the shifts between community assemblages at different depths. Pearson correlations ($r = \pm0.4$) indicated that ostracods,

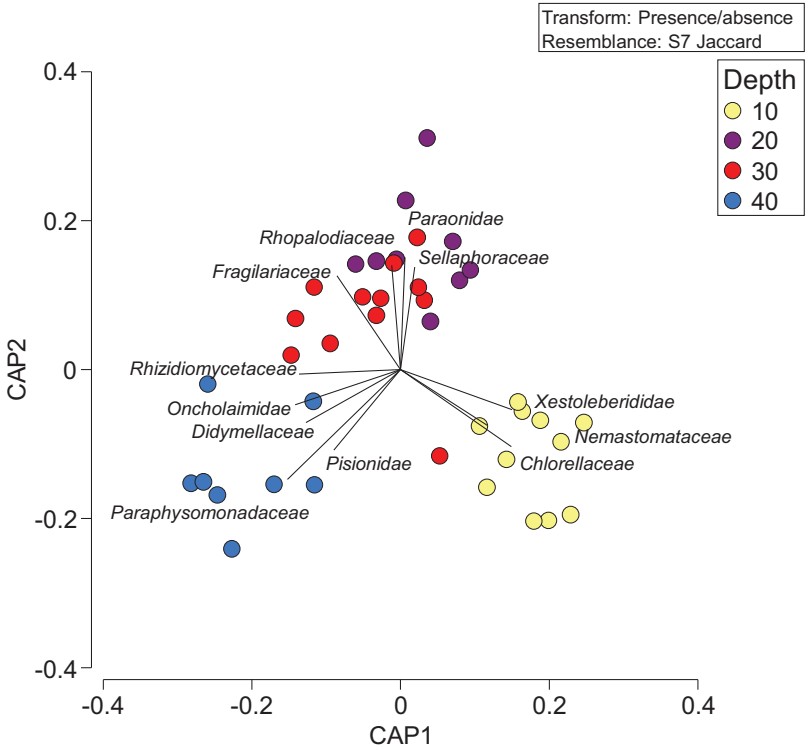

**Figure 4 Presence/absence of eukaryotic families collected at two coral reefs in Okinawa, Japan.** Constrained Canonical Analysis of Principal Coordinates (CAP) comparing presence/absence of eukaryotic families detected based on sediment samples collected at two coral reefs in Okinawa, Japan and 18S rRNA sequences. The relationship of eukaryotic community assemblages identified in each sample using a Jaccard resemblance matrix for the factor "depth" is shown, with different depths indicated by colors in the legend. Pearson correlation vectors ($r > 0.4$) represent the eukaryotic taxa driving the relationship among samples.

nematodes, polychaete worms, fungi, and marine algae and diatoms were the taxa most closely correlated with distinct depths. Green (Chlorellaceae) and red algae (Nemastomataceae) as well as ostracods (Xestoleberididae) were associated with 10 m, polychaetes (Paraonidae) and diatoms (Rhopalodiaceae, Fragilariaceae) were associated with 20 and 30 m, and polychaetes (Pisionidae), nematodes (Oncholaimidae), fungi (Didymellaceae), and chrysophyte algae (Paraphysomonadaceae) were associated with 40 m (Fig. 4).

The depth zonation apparent with taxonomy-dependent approaches was supported by the comparison of OTUs across depths within the combined data set including classes Anthozoa and Demospongiae (Fig. 5; Table S3). PERMANOVA tests indicated significant differences between depths ($p = 0.046$, d$f = 3$, pseudo-F = 1.3). Pearson correlations ($r = \pm0.4$) indicated that OTUs from the class Demospongiae (and not Anthozoa) were most closely correlated with different depths (OTU12, OTU27, OTU44, OTU45, and OTU125), suggesting that sponges, and perhaps not anthozoans/corals, may be better indicators of depth given their greater relative read abundance and fine-scale zonation (Fig. 5). OTU12 and OTU27, which were correlated with the shallowest depth (10 m), represent species within Haploscleromorpha clade E and Astrophorina

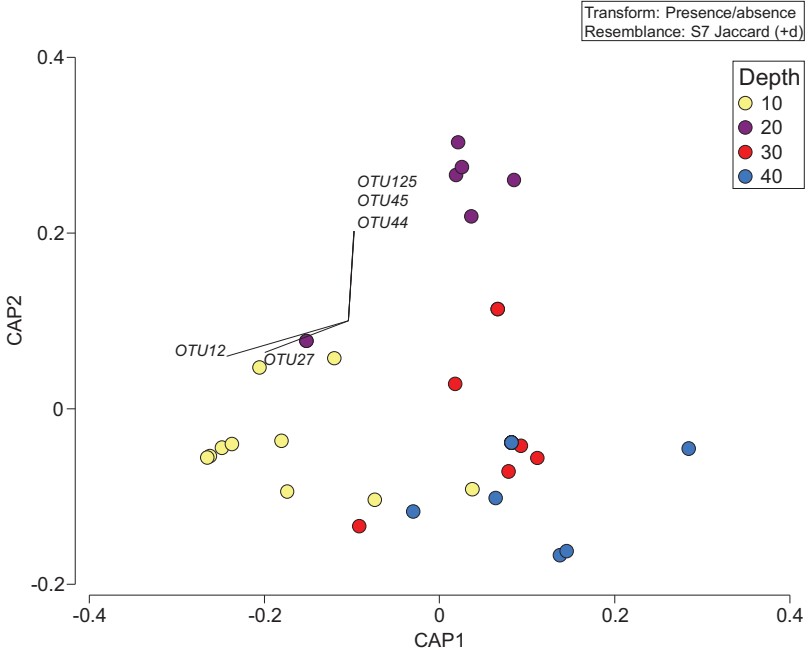

**Figure 5 Presence/absence of the combined OTU dataset for class Anthozoan and Demospongiaecollected at two coral reefs in Okinawa, Japan.** Canonical Analysis of Principle Coordinates (CAP) ordination plot of the presence/absence of the combined OTU dataset for class Anthozoan and Demospongiae based on sediment samples collected at two coral reefs in Okinawa, Japan and 18S rRNA sequences. The relationship of OTUs identified in each sample using a Jaccard resemblance matrix for factor "depth" is shown, with different depths indicated by colors in the legend. Pearson correlation vectors ($r > ±0.4$) represent the OTUs driving the relationship among samples; all of these OTUs are from the class Demospongiae.

(see *Redmond et al., 2013*), boring or encrusting and carbonate reef associated sponges, respectively. OTU44, OTU45, and OTU125, on the other hand, which appear to be correlated with 20 m depth, represent species within Haplloscleromorpha clade C (OTU44 and OTU45) and Poecilosclerida (OTU125).

# DISCUSSION

The RUMS eDNA approach utilized in this pilot study may be suited to tracking changes in biodiversity across small spatial and temporal scales, as evidenced by the wide spectrum of biodiversity obtained at each site and the consistent grouping of replicate samples (irrespective of reef or year) by depth (Figs. 4 and 5). Previous work has shown that biotic composition characterized by eDNA differs between depths of 0 and 20 m or 40 m in Monterey Bay (*Andruszkiewicz et al., 2017*), and between sites separated by 75–4,000 m at the same depth (*O'Donnell et al., 2017*). Our sediment metabarcoding results demonstrate even finer scale resolution, with notable and significant differences in marine community assemblages at coral reef sites separated by 10 m depth and less than 240 m total distance based on a 45 degree reef slope. Collectively, these studies indicate that there are spatial patterns in the organization of eDNA in marine sediments and that it is not homogenous. Based on the null results for the partitioning of beta-diversity (i.e., family

richness) among depths (Fig. 3A), we suggest that the substitution of species may be due to competition, environmental filtering, or historical events that made the highest relative contribution (also see *Pearman et al., 2018*) to the fraction of biodiversity that we sequenced. We therefore focus on significant shifts in eukaryotic community assemblages with depth in the remainder of the discussion.

Although we detected numerous eukaryotic taxa at the family level with our RUMS, these results likely only reflect a fraction of the total biodiversity present in the immediate environment due to biases introduced by using different sampling substrates (*Koziol et al., in press*), using a single metabarcoding assay (*Stat et al., 2017*), and the limited availability of genetic reference sequences (*Chain et al., 2016*). For example, with regards to metabarcoding assays, recent data suggest that the use of multiple primer sets, as opposed to a single universal PCR assay, can identify a greater richness of marine biodiversity of a given site or sample (*Kelly et al., 2017*; *Stat et al., 2017*). Indeed, single DNA marker assays suffer from primer bias (thus excluding entire taxonomic groups), PCR or sequencing artefacts, low taxonomic resolution, and contamination issues (*Schloss, Gevers & Westcott, 2011*), although the impact of these effects depend on whether you assay and compare relative vs. absolute biodiversity.

Our study, like others, highlights the impact of incomplete reference DNA databases for many marine taxa across loci that are easily targeted by metabarcoding—on average two-thirds of our metabarcodes could not be assigned with fidelity at the family level following QF and querying against NCBI GenBank, the largest open access, annotated collection of nucleotide sequences in the world. This is not surprising given that members of the phyla Nematoda and Platyhelminthes, which make up a significant fraction of the marine biodiversity in sedimentary material, particularly in deep oceanic environments, are often the most poorly characterized genetically (*Sinniger et al., 2016*). Similarly, the large majority of our Demospongiae 18S sequences matched those vouchered in a single publication (*Redmond et al., 2013*). Based on this it is clear that more comprehensive DNA sequence reference databases are needed, particularly for understudied or cryptic invertebrate groups. For Anthozoa in particular, it should be noted that we did not detect any Octocorallia within our dataset despite the relative commonality and high diversity of this group on coral reefs in Okinawa (*Lau et al., 2018*). There is a relatively large 18S rRNA dataset on GenBank for this group (>980 sequences as of November 26, 2018), and thus we attribute our results to low cellular shedding rates, limitations of the assay, or the fact that these targets are not present in high concentrations in sediment (see *Koziol et al., in press*). Despite these limitations, the similarities of taxonomy-dependent community assemblages between replicates at the same depth in our study are striking. Although shifts in community assemblages as little as 10 m apart may initially seem surprising, biotic differences in flora and fauna across small changes in depth are well known from coral reefs (*Friedlander & Parrish, 1998*; *Kahng & Kelley, 2007*; *Brokovich et al., 2008*), and the eDNA in our study reflects such patterns at least to a degree that is statistically significant (Fig. 4).

With these caveats in mind, when time and money are limited, and the goal of the study is a comparison among samples or sites vs. identifying the entire marine tree of life

in the environment, the extra effort and expenditure may not even be warranted. For example, *Stat et al. (2017)* demonstrated that PCR assays based on the commonly employed 18S rDNA V4 region detected the greatest proportion of taxa (44% of the total number of families) among the ten total PCR assays examined (also see *Kelly et al., 2017*). Moreover, RUMS provide information on a subset of benthic organisms or the DNA of planktonic organisms that settle and accumulate in the sediment, and not the entire marine tree of life. *Pearman et al. (2018)* detected higher biodiversity with multiple primers but showed that similar patterns were found when comparing the two different primer sets. Thus, depending on the goal(s) of the study, expanding to other substrates, assays, or improving the underlying taxonomic assignments may be advantageous.

In this study, we attempted to overcome the lack of reference databases by performing additional taxonomic-independent approaches (e.g., OTU analyses) on two important classes or organisms associated with coral reefs, Anthozoa and Demospongiae. These analyses revealed that demosponge DNA was more common in RUMS, and also more helpful in discriminating between depths on a fine-scale (Fig. 4). Even with this approach, robust identification of many Demospongiae OTUs to species or genus level still remained problematic. Again, this is due to the large amount of taxonomic work that remains to be done in this group (*Van Soest et al., 2012*; *Redmond et al., 2013*). As a result, our taxonomic assignment of OTUs was limited to large molecular clades at the suborder/order level. An additional limitation is related to specimen discovery; sponges are often cryptic on reefs, and include boring or encrusting species that can adhere to the undersides of rocks and coral rubble, or live inside the coral carbonate matrix, making post-survey ground-truthing difficult.

Our eDNA metabarcoding data was able to generate a set of OTUs that could potentially be used as indicators for different depths. This result is important as it provides targets for future morphological studies and will also help refine metabarcoding assays to better qualify select taxa. In this data, OTUs 12, 27, 44, 45, and 125 stood out as key discriminating taxa at Cape Hedo and Rukan. OTU12 (unidentified Haploscleromorpha clade E species; *sensu Redmond et al., 2013*) and OTU27 (*Penares* sp.), detected in 12.5% and 5% of the replicates, respectively, primarily from sediment sampled at 10 m depth from both sites, represent a mixture of boring or encrusting and carbonate reef associated sponges. These taxonomic groups might therefore be good indicators in coral reef-associated areas. OTU44, OTU45, and OTU125, on the other hand, were based on rare detections (2.5% of the replicates in each case) at Cape Hedo, and only at 20 m depth. Most of these taxa are in groups known from coral reefs in Japan, and Okinawa-jima in particular. Indeed, the taxonomic group corresponding to OTU44 and OTU45 (Haploscleromorpha clade C; *sensu Redmond et al., 2013*) are a source of manzamines, a polycyclic alkaloid with anti-microbial and anti-leukemic properties that were initially discovered and described from a site on the west coast of Okinawa-jima (Cape Manza; *Sakai et al., 1986*). OTU125 was an unidentified Poecilosclerida species, with no close matches in GenBank (i.e., closest match cf. *Hymedesmia* sp., 375 out of 392 bp matching).

## CONCLUSIONS

In the context of a rapidly warming ocean and eutrophication of coastal environments, effective biodiversity monitoring is vital to understanding and predicting how the taxonomic composition of coral reef ecosystems might change. Importantly, these kinds of eDNA data will provide an evidence base to develop appropriate management plans. Given the patterns observed in this data, future RUMS would be well-served to examine even finer scale differences on coral reefs, including expansion of eDNA surveys to other sites and across multiple seasons/years. Taken together, this study adds to a growing body of evidence that eDNA metabarcoding, even in its current state of development, represents a powerful way to explore marine biodiversity across environments. The proportion of RUMS data that remains without taxonomic assignment also brings into focus the need for more complete DNA reference databases underpinned with a robust taxonomy. An integrative framework of eDNA and more classical (morphology-based) taxonomy are needed, in tandem, to characterize marine taxa that sit at the base of the marine food web in coral reef ecosystems.

## ACKNOWLEDGEMENTS

The authors would like to acknowledge Matthew Power and Megan Coghlan for DNA sequencing assistance. In Okinawa, we thank Yoshihiro Katsushima and the Rukan boat captain for field work assistance, as well as members of the MISE Laboratory at the University of the Ryukyus.

### Funding

This study was funded by the Pawsey Supercomputing Centre, the Australian Research Council (LP160100839 and LP16101508), a Joint Usage and Collaborative Research Grant from the Tropical Biosphere Research Center (TBRC) at the University of the Ryukyus to Joseph D. DiBattista and James D. Reimer, as well as a Curtin University Early Career Research Fellowship (ECRF) to Joseph D. DiBattista and an Environment and Agriculture Visiting Scholarship to James D. Reimer. The funders had no role in study design, data collection and analysis, decision to publish, or preparation of the manuscript.

### Grant Disclosures

The following grant information was disclosed by the authors:
Pawsey Supercomputing Centre, the Australian Research Council: LP160100839 and LP16101508.
Tropical Biosphere Research Center (TBRC) at the University of the Ryukyus.
Curtin University Early Career Research Fellowship: ECRF.
Environment and Agriculture Visiting Scholarship.

### Competing Interests

James D. Reimer is an Academic Editor for PeerJ.

## Author Contributions

- Joseph D. DiBattista conceived and designed the experiments, performed the experiments, analyzed the data, contributed reagents/materials/analysis tools, prepared figures and/or tables, authored or reviewed drafts of the paper, approved the final draft.
- James D. Reimer conceived and designed the experiments, performed the experiments, analyzed the data, contributed reagents/materials/analysis tools, prepared figures and/or tables, authored or reviewed drafts of the paper, approved the final draft.
- Michael Stat performed the experiments, analyzed the data, contributed reagents/materials/analysis tools, prepared figures and/or tables, authored or reviewed drafts of the paper, approved the final draft.
- Giovanni D. Masucci performed the experiments, prepared figures and/or tables, authored or reviewed drafts of the paper, approved the final draft.
- Piera Biondi performed the experiments, prepared figures and/or tables, authored or reviewed drafts of the paper, approved the final draft.
- Maarten De Brauwer analyzed the data, contributed reagents/materials/analysis tools, prepared figures and/or tables, authored or reviewed drafts of the paper, approved the final draft.
- Michael Bunce conceived and designed the experiments, performed the experiments, contributed reagents/materials/analysis tools, authored or reviewed drafts of the paper, approved the final draft.

## Data Availability

Data available from the Dryad Digital Repository: https://doi.org/10.5061/dryad.37qv5rd.

DiBattista, Joseph; Davis Reimer, James; Stat, Michael; Masucci, Giovanni; Biondi, Piera; De Brauwer, Maarten; et al. (2019): Raw .fastq sequence files for PeerJ submission "Digging for DNA at depth: rapid universal metabarcoding surveys (RUMS) as a tool to detect coral reef biodiversity across a depth gradient." figshare. Fileset. https://doi.org/10.6084/m9.figshare.7453172.v1.

## Supplemental Information

Supplemental information for this article can be found online at http://dx.doi.org/10.7717/peerj.6379#supplemental-information.

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
