# Peer review of "Digging for DNA at depth: rapid universal metabarcoding surveys (RUMS) as a tool to detect coral reef biodiversity across a depth gradient"

_PeerJ, doi:10.7717/peerj.6379_

## Round 0.1 · original submission · Major Revisions

Dear Joseph,

I have received assessments from two contrasting reviewers. Both reviewers acknowledged that the study is well written, the study site interesting and the data informative. Reviewer #1 had only minor comments. Reviewer #2 expressed some concerns about the study design and advised to remove the two sites comparison and only report on the overall diversity. In your rebuttal, I will expect you to carefully address this concern, and will be looking forward receiving the revised manuscript.

With regards,
Xavier

·

Basic reporting

The English used throughout the manuscript is clear and unambiguous. There are small points which require attention. See below.

Line 87 -88 – Two sets of brackets present which could be combined together I think.

Line 97 – 99 – While the authors are correct with their quotation Taberlet et al 2018 in his recent book (“Environmental DNA: For Biodiversity Research and Monitoring”) have established a more comprehensive definition of eDNA that I believe should at least be considered by the authors although their current definition would just fall into the extracellular eDNA classification.

Line 104: This is where I believe the clear definition of eDNA is required as eDNA samples targeting protists will have an obvious sign of biological material (even if a microscope is required) and thus would not in a strict sense be included in your classification of eDNA. While personally I do not disagree with the authors about the inclusion of the protists studies as examples of eDNA I think that the definition just has to be clarified.

L113: Should include that this is partially due to the lack of barcoded reference organisms in particular.

L139: I believe as this is the first time as far as I can see you use the acronym RUMS in the main text, you require the word rapid before universal metabarcoding surveys


Line 223: do not require the word those

Line 233 – 238: Could you please shorten the sentence somewhat.


Line 340: Delete one of the “that” and replace with the

Line 343: what is the 1m in the brackets for.



Figure 1: While the figure is easy to understand I have a small issue with the arrows and the one for light especially. It has the shading at the surface and getting lighter with depth. I admit this is just a personal opinion but to me it would be more logical to be light at the surface and get darker with depth as happens in the water column. If they authors did this to match the other two arrows (nutrients and temperature) then maybe using different colours would improve …e.g. shades of blue for light and red for temperature.

Figure 2: The contrast colors – what are the segments within other – is it just smaller phyla?

Experimental design

Line 161: How was the sediment exactly obtained. Was it just scooped up with the falcon tube?

Line 170: I am slightly confused with the 4 replicate samples. On line 161 you mention 2 replicates. Could you please just clarify this.

Line 179: Please see previous comment about replicates. If the number of replicates per site/depth was two of such a small amount of sediment I would be dubious about the reliability of the results. If it is 4 replicates this is better.


L249: What were the BLASTn settings used?

Line 272: What packages with R were used?

Validity of the findings

Line 360: Indeed multiple primers can detect higher diversity but Pearman et al 2018 showed that similar patterns were found using two different primer sets. So it depends on what the aims of the study are. If you just want to find all species then multiple primers would be necessary but in this study if you just wanted to detect changes then maybe only one primer is required. This should be taken into account especial when discussing time and money later on

Additional comments

This is generally a well written article examining the use of eDNA to detect differences in diversity along a depth gradient in two coral reefs in Japan. The study appears robust with just a few clarifications required in the methods section to clarify some of the methodologies undertaken.

Reviewer 2 ·

Basic reporting

The writing and grammar are fine. References seem in order for the field of marine eDNA, though there is a lack of discussion/referencing for expected marine ecology. Figures are professional, however the statistical analysis should be treated with caution, especially when comparing between sites due to the experimental design and extraction methods.

Experimental design

The findings are a bit of pilot data that should be treated as such. Overall, the data sampling is unbalanced and lacks spatial structure. Compounded with single 0.5 gram eDNA extractions the data derived from the samples should be interpreted with caution due to the random error associated with the experimental design. In general, drop the statistical analyses comparing between sites and report the overall diversity with some discussion to the expected diversity in the system. Drawing too much from beta-diversity measures is likely to be misleading.

Validity of the findings

see section 2

---

## Round 0.2 · accepted · Accept

Dear Joseph and co-authors,

Thank you for your revised manuscript. I am pleased with the way you have addressed all reviewers comments and modified the manuscript accordingly. I am delighted to accept this work for publication in PeerJ - thank you for the nice contribution to the field!

In the proofs, please do not forget to include the correct Dryad Digital
Repository accession # for your raw data.

Happy New Year!
Xavier

# ·

Basic reporting

The paper is well written

Experimental design

The authors have addressed the issues that I had previously pointed out

Validity of the findings

no comment